# Signaling diversity enabled by Rap1-regulated plasma membrane ERK with distinct temporal dynamics

Jeremiah Keyes[1], Ambhighainath Ganesan[2†], Olivia Molinar-Inglis[1], Archer Hamidzadeh[3], Jinfan Zhang[1], Megan Ling[4], JoAnn Trejo[1], Andre Levchenko[3], Jin Zhang[1,4,5]*

[1]Department of Pharmacology, University of California San Diego, La Jolla, United States; [2]Department of Pharmacology and Molecular Sciences, The Johns Hopkins University School of Medicine, Baltimore, United States; [3]Department of Biomedical Engineering and Yale Systems Biology Institute, Yale University, New Haven, United States; [4]Department of Chemistry and Biochemistry, University of California San Diego, La Jolla, United States; [5]Department of Bioengineering, University of California San Diego, La Jolla, United States

**Abstract** A variety of different signals induce specific responses through a common, extracellular-signal regulated kinase (ERK)-dependent cascade. It has been suggested that signaling specificity can be achieved through precise temporal regulation of ERK activity. Given the wide distrubtion of ERK susbtrates across different subcellular compartments, it is important to understand how ERK activity is temporally regulated at specific subcellular locations. To address this question, we have expanded the toolbox of Förster Resonance Energy Transfer (FRET)-based ERK biosensors by creating a series of improved biosensors targeted to various subcellular regions via sequence specific motifs to measure spatiotemporal changes in ERK activity. Using these sensors, we showed that EGF induces sustained ERK activity near the plasma membrane in sharp contrast to the transient activity observed in the cytoplasm and nucleus. Furthermore, EGF-induced plasma membrane ERK activity involves Rap1, a noncanonical activator, and controls cell morphology and EGF-induced membrane protrusion dynamics. Our work strongly supports that spatial and temporal regulation of ERK activity is integrated to control signaling specificity from a single extracellular signal to multiple cellular processes.

*For correspondence:
jzhang32@ucsd.edu

Present address: †IBM, Global Business Services, New York, United States

Competing interests: The authors declare that no competing interests exist.

## Introduction

Due to its central role in signal transduction, extracellular-signal regulated kinase (ERK) has been the subject of intense study for over three decades (*Seger et al., 1991*; *Seger and Krebs, 1995*); this research has elucidated many of the key mechanisms of ERK regulation. This pathway is utilized by a host of different extracellular cues to regulate cellular processes such as proliferation, differentiation, and survival (*Keshet and Seger, 2010*; *Roskoski, 2012*). However, studies have shown that ERK is also critically involved in regulating many other cellular processes, such as cell migration (*Mendoza et al., 2015*; *Mendoza et al., 2011*), cell cycle progression (*Roberts et al., 2006*; *Chambard et al., 2007*), autophagy (*Hui et al., 2016*; *Jo et al., 2014*), metabolism (*Shin et al., 2015*; *Jiao et al., 2013*), insulin secretion (*Ozaki et al., 2016*), or even apoptosis (*Li, 2014*; *Cagnol and Chambard, 2010*). The variety of processes induced through the ERK pathway presents a conundrum: how do extracellular cues specifically regulate the ERK pathway to induce the proper cellular response?

The canonical mode of regulation is activation by phosphorylation at the end of a kinase cascade. ERK is phosphorylated by the protein kinase mitogen-activated protein kinase kinase (MEK), which is first activated by the protein kinase Raf. Raf kinases are activated through the GTPase Ras, which is activated downstream of extracellular signals such as Epidermal Growth Factor (EGF) (*Roskoski, 2012*; *Canagarajah et al., 1997*). However, because ERK is activated by phosphorylation by a host of divergent extracellular signals, more complex modes of regulation must exist in order to direct ERK activity toward appropriate responses.

Previous studies have shown that different extracellular signals induce distinct temporal patterns of ERK activation to encode specific signaling information (*Murphy et al., 2002*; *Herrero et al., 2016*; *Sasagawa et al., 2005*). For example, in model cell lines such as PC12 or MCF7, EGF treatment leads to transient ERK phosphorylation and cell proliferation, whereas alternate extracellular signals such as Nerve Growth Factor (NGF) or heregulin, respectively, induces sustained ERK phosphorylation and cell differentiation. In the case of PC12 cells, the latter includes neurite formation and an enlarged cell body, modeling differentiated neurons. However, previous studies primarily used immunoblotting techniques to monitor ERK activation dynamics. While this technique has yielded crucial information, it is ultimately limited in its spatiotemporal resolution and unable to resolve single-cell behavior or differential dynamics between subcellular compartments.

ERK substrates have been identified in many subcellular compartments including at the Golgi Apparatus (*Jesch et al., 2001*; *Shaul and Seger, 2006*; *Wainstein and Seger, 2016*), at the mitochondrial membrane (*Tomer et al., 2018*; *Nagdas et al., 2019*), and near the plasma membrane (*Mendoza et al., 2015*; *Mendoza et al., 2011*). However, studies investigating the spatial regulation of ERK have primarily focused on understanding its activation and role in shuttling between the nucleus and cytoplasm, leaving the regulation and function of ERK in other subcellular compartments relatively unexplored. We hypothesize that spatial compartmentalization of ERK activity is required for signaling specificity in order for signals such as EGF to regulate different functional responses via distinct compartmentalized ERK activity. To test our hypothesis, we utilized genetically encoded molecular tools to visualize and perturb the activities of specific subcellular pools of endogenous ERK. Using this native biochemistry approach (*Sample et al., 2014*; *Mehta and Zhang, 2017*; *Lin et al., 2019*), we discovered that ERK activity associated with the plasma membrane exhibits distinct temporal dynamics in contrast to cytoplasmic and nuclear pools of ERK. We further dissected the mechanism of this distinct regulation and discovered that plasma membrane ERK activity is required for regulating Rac1 and controlling actin protrusion dynamics.

## Results

### EGF-induced ERK activity is transient in the cytoplasm but sustained near the plasma membrane

While previous studies revealed that the temporal dynamics of ERK activation play a pivotal role in promoting specific cellular responses to extracellular stimuli (*Murphy et al., 2002*; *Herrero et al., 2016*; *Herbst et al., 2011*), studies on the spatial activation of ERK in signal transduction has been mostly focused on its translocation to the nucleus despite the fact that ERK has hundreds of substrates in other subcellular compartments (*Yoon and Seger, 2006*). For example, ERK has several known substrates near the plasma membrane, including Focal Adhesion Kinase (*Jiang et al., 2007*) and the actin-effector Wiskott–Aldrich Syndrome Protein (WASP) (*Mendoza et al., 2011*). Thus, we sought to determine whether a single extracellular signal could induce different activity dynamics of spatially distinct pools of ERK to promote functional cellular processes. To address this question, we developed a new toolset of genetically encoded, spatially localized fluorescent biosensors based on the ERK Kinase Activity Reporter (EKAR) (*Herbst et al., 2011*; *Komatsu et al., 2011*), an engineered fluorescent ERK substrate that will exhibit different FRET properties in response to phosphorylation by ERK. These biosensors consist of two fluorescent proteins flanking a WW phosphopeptide-binding domain, a linker region, and an ERK-specific substrate domain (*Figure 1A*). Upon phosphorylation of the substrate sequence by ERK, the WW phosphopeptide-binding domain interacts with the phosphorylated substrate sequence, bringing the two fluorescent proteins into close proximity, thereby increasing FRET (*Stryer, 1978*). Thus, by expressing the biosensor within cells, one can monitor ERK activity in living cells in real-time by exciting the donor fluorescent protein and measuring

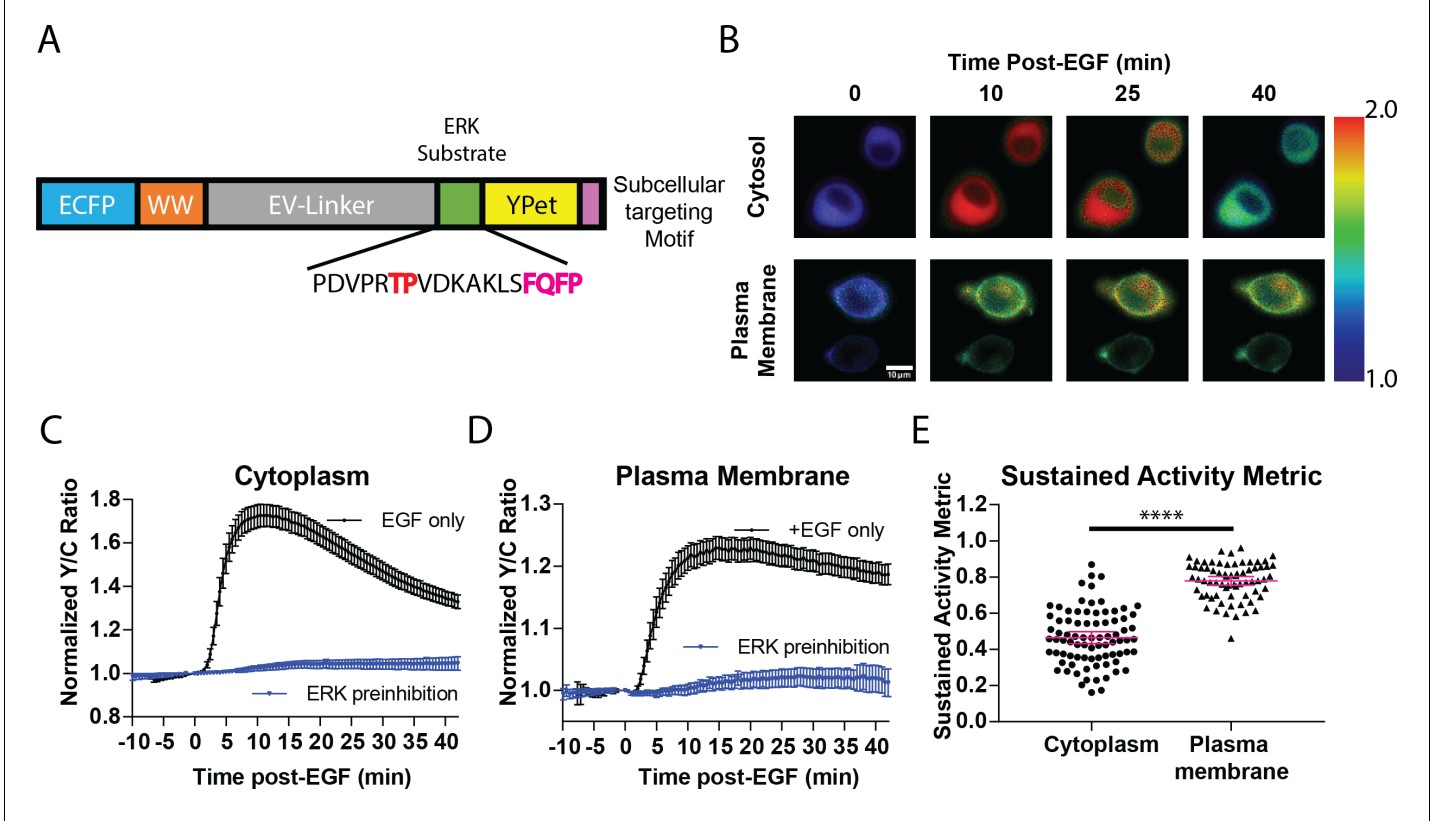

**Figure 1.** Targeted, FRET-based ERK biosensors reveal differential temporal dynamics of subcellular ERK activity. (A) Domain structure of improved ERK-kinase activity reporter (EKAR4). EKAR4 has two fluorescent proteins (ECFP and YPet) on N- and C-termini, respectively, with an ERK-specific substrate sequence, a phosphopeptide-binding domain (WW), and the EV-linker developed by *Komatsu et al., 2011* (B) Pseudocolor images representing the yellow over cyan (Y/C) emission ratio of cytosolic EKAR4 (cytoEKAR4) (top) versus plasma membrane-targeted EKAR4 (bottom) after EGF treatment at time 0. Warmer colors indicate higher Y/C emission ratio, scale bar = 10 μm. (C) Spatiotemporal dynamics of EGF-induced ERK activity in the cytosol. Cytoplasmic ERK activity was monitored using cytoEKAR4 in cells treated with 100 ng/μl EGF (black dotted line, n = 83) or pretreated with 10 μM SCH772984 (ERK inhibitor) 20 min before EGF (blue, triangles, n = 19). Each trace is a combined average of all cells. (D) Spatiotemporal dynamics of EGF-induced ERK activity at the plasma membrane. Plasma membrane localized ERK activity was monitored using the plasma membrane targeted EKAR4 (pmEKAR4) in cells treated with 100 ng/μl EGF (black dotted line, n = 71) or pretreated with 10 μM SCH772984 (ERK inhibitor) 20 min before EGF (blue, triangles, n = 24). Each trace is a combined average of all cells (see *Figure 1—figure supplement 1A*, B for traces of all cells for C and D; Error bars represent 95% CI.) E) Activity persistence differences of ERK response to EGF between the cytoplasm and plasma membrane. Using the SAM40 metric (*Equation 1*), the transient versus sustained nature of ERK response to EGF at the cytoplasm (n = 83) and plasma membrane (n = 71) was quantified. (****p<0.0001 using one-way ANOVA multiple comparisons, see *Figure 1—figure supplement 1D* for comparison to nuclear ERK activity.) See also *Figure 1—figure supplement 1*, *Figure 1—figure supplement 2*.

The online version of this article includes the following source data and figure supplement(s) for figure 1:

**Source data 1.**

**Figure supplement 1.** Direct comparison of EKAR4 to previous generations of EKAR in HEK-293T cells.

**Figure supplement 2.** Verification of biosensor localization.

**Figure supplement 3.** Single cell traces of EKAR responses reveal differential dynamics between subcellular locations.

**Figure supplement 4.** No significant correlation between EKAR4 signal amplitude and persistence.

**Figure supplement 5.** phoshpo-ERK time course.

the changes in the ratio of acceptor to donor fluorescence, yellow over cyan (Y/C), which is often reported as a percent increase over basal signal. Previous versions of EKAR include EKAR-EV, EKAR2G, and EKAR3 (*Komatsu et al., 2011*; *Fritz et al., 2013*; *Sparta et al., 2015*). While each of these biosensors has improved upon past biosensors, each of their reported average response to EGF is below 50%, which limits their utility to be used when targeted to specific subcellular locations. In seeking to develop a new EKAR biosensor with sufficient dynamic range in order to maximize signal when the biosensor is restrained to specific locations, we found that switching the

positions of the fluorescent proteins within the EKAR-EV biosensor such that Ypet was positioned C-terminally and ECFP was positioned N-terminally (*Figure 1A*) yielded a biosensor that generated an 74.5 ± 2.3% increase in the Y/C emission ratio in PC-12 cells (n = 83) (*Figure 1—figure supplement 1*). We also directly compared EKAR4 to EKAR2G1 and EKAR-EV in HEK-293 cells and observed a significantly improved dynamic range from the previous biosensors (*Figure 1—figure supplement 1*). We reasoned that this enhanced EKAR, which we have termed EKAR4, can then be targeted to specific subcellular locations, such as the cytoplasm, plasma membrane, or nucleus, to enable reliable tracking of ERK temporal dynamics in these specific locations. Thus, we used a Nuclear export signal (NES) to create a cytosolic targeted EKAR4 (cytoEKAR4), a nuclear localization sequence (NLS) to create a nuclear targeted EKAR4 (nuclearEKAR4), and the hypervariable region of KRas including a polylysine and CAAX box sequence to create a plasma membrane-localized EKAR4 (pmEKAR4), each of which displayed correct localization (*Herbst et al., 2009*; *Figure 1B*, *Figure 1—figure supplement 2*).

Live-cell imaging experiments revealed striking differences in the temporal patterns of ERK activity at different subcellular locations. EGF induced transient ERK activity in the cytosol and nucleus (*Figure 1C*, *Figure 1—figure supplement 3B,C*), consistent with previous reports. In contrast, EGF-stimulated pmEKAR4 response was much more sustained at the plasma membrane (*Figure 1D*, *Figure 1—figure supplement 3A*), compared to cytosol and nucleus. We developed a metric to quantitate the relative temporal dynamics between conditions (*Equation 1*), which we refer to as Sustained Activity Metric at 40 min post-treatment (SAM40), where *R* represents the normalized emission ratio at the indicated condition in subscript.

$$\frac{R_{40} - R_0}{R_{max} - R_0} \tag{1}$$

An advantage of this metric is that it presents the residual signal at 40 min from the maximum signal after treatment. Using this metric, we were able to determine that EKAR4 response at the plasma membrane was indeed statistically different from the cytoplasmic and nuclear ERK responses (*Figure 1E*, *Figure 1—figure supplement 3D*, *Figure 1—figure supplement 4*). Interestingly, we also found that the kinetics of ERK activity accumulation were slower at the plasma membrane versus cytosol and nucleus (*Figure 1—figure supplement 3F*), using the time to ½ maximum responses as a proxy (*Bünemann et al., 2003*; *Ryu et al., 2015*; *Wan et al., 2019*), which is dependent on both ERK and phosphatase activity.

We also compared the EKAR4 responses to conventional methods of evaluating ERK by treating PC-12 cells with EGF and harvesting cells before treatment (control), at 7.5 min, 15 min, 30 min, and 40 min post-EGF treatment (*Figure 1—figure supplement 5*) and subjected cell lysates to immunoblot for phosphorylated and total ERK. In this time course, we observe a dramatic increase in phosphorylated ERK 7.5 min after EGF treatment, followed by a dramatic decrease by 15 min. It is interesting that after 15 min we see a slight increase in phospho-ERK at the 30- and 40 min time points. The activation profile is broadly similar to the EKAR response. The decrease in phospho-ERK is faster than the EKAR signal, likely because the EKAR biosensor signal dynamics is dependent on both ERK and phosphatase activities.

To test if the sustained response at the plasma membrane requires continuously active ERK at the plasma membrane, we treated the cells with a membrane permeable ERK inhibitor SCH772984 (*Morris et al., 2013*) and examined the pmEKAR4 responses. We found that addition of 10 μM SCH772984, but not of DMSO (*Figure 2—figure supplement 1C,D*), either at 10 min after EGF stimulation (*Figure 2A* blue curve) or at 40 min post-stimulation (*Figure 2A* orange curve) resulted in an immediate change in the slope of pmEKAR4 signal, indicating that ERK was still actively phosphorylating pmEKAR4 at these time points (*Figure 2A*, *Figure 2—figure supplement 1*). Interestingly, inhibitor addition at these different time points appears to exhibit different slopes of decay, which we surmise to be due to changing phosphatase activities between these time points. Furthermore, addition of U0126, an inhibitor of the upstream kinase MEK, at 10 min or 40 min post EGF also led to decreases in the slope of pmEKAR4 signal (*Figure 2—figure supplement 2*), indicating MEK was also active.

To further validate the ERK activity dynamics at different subcellular compartments, we performed biochemical subcellular fractionation (*Figure 2—figure supplement 3*) to separate and quantitate ERK phosphorylation at the plasma membrane and cytoplasm of PC12 cells to assess the

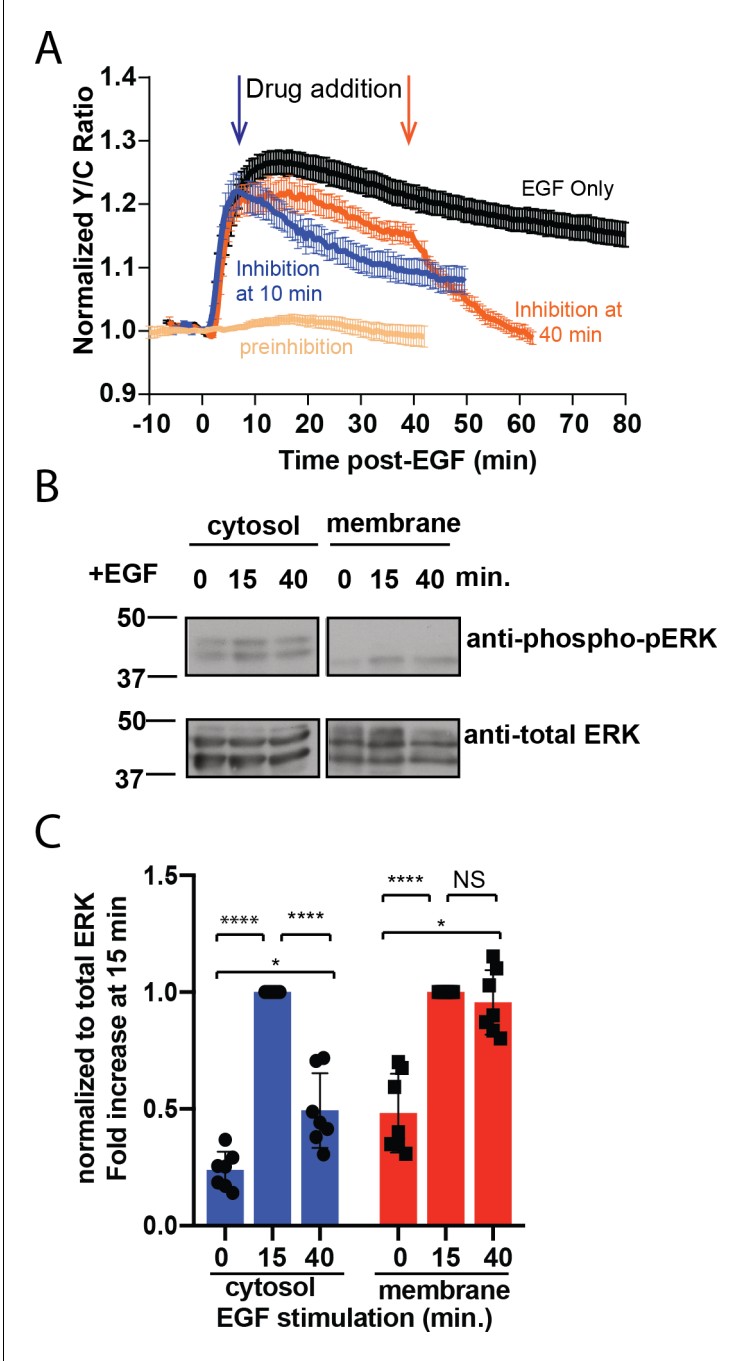

**Figure 2.** Sustained ERK activity at the plasma membrane is required for observed pmEKAR4 signal. (**A**) PC12 cells treated with the ERK inhibitor SCH772984 (10 µM) after EGF treatment at 10 min (n = 8) or 40 min (n = 8) post-EGF resulted in an immediate change in the slope of pmEKAR4 signal. (**B**) PC12 cells were harvested at select time points after EGF treatment and the lysates underwent subcellular fractionation, which is verified in *Figure 2— figure supplement 1E*. After successful fractionation between the plasma membrane and other cellular components, a western blot against the phosphorylated and total form of ERK1/2 indicates that the levels of phospho-ERK remain relatively consistent up to 40 min after EGF treatment. Quantitation from five independent replicates is shown in panel C. (*p<0.05, ****p<0.0001, calculated using one-way ANOVA with multiple comparisons.) See also *Figure 2—figure supplement 1*.

The online version of this article includes the following source data and figure supplement(s) for figure 2:

**Source data 1.**

**Figure supplement 1.** Responses of pmEKAR4 with ERK inhibitor treatment.

*Figure 2 continued on next page*

*Figure 2 continued*

**Figure supplement 2.** Reponses of pmEKAR4 with MEK inhibitor treatment.
**Figure supplement 3.** Verification of successful subcellular fractionation.

temporal dynamics of phosphorylated ERK. ERK was phosphorylated transiently in the cytoplasm and ERK associated with the plasma membrane was, on average, equally phosphorylated between the 15- and 40 min time points (*Figure 2B*, quantitated from five independent replicates in C), indicating the presence of a continuously phosphorylated pool of ERK at the plasma membrane. In conclusion, these data indicate that ERK activity associated with the membrane is sustained in contrast to the transient response in the cytoplasm and nucleus, and thus differentially regulated between these subcellular compartments.

## Rap1 GTPase significantly contributes to plasma membrane ERK activity

The unique temporal pattern of ERK activity at the plasma membrane prompted us to determine whether plasma membrane ERK activity is distinctly regulated. The classical pathway of ERK activation is through the Ras family of GTPases, including K-ras, H-ras, and N-Ras (*Figure 3—figure supplement 1A*). However, it has been shown that Rap1, a GTPase in the Ras family, can in some cases specifically interact with B-Raf and lead to sustained ERK activity under certain conditions (*Takahashi et al., 2017*; *Li et al., 2016*; *York et al., 1998*). For example, NGF-induced sustained ERK activation in PC12 cells is mediated through Rap1 (*York et al., 1998*). More recent reports suggest that while Ras activates a rapid and intense ERK response, Rap1 induces a slow and steady ERK response (*Li et al., 2016*), mimicking the observed kinetic difference between responses of pmE-KAR4 and cytoEKAR4. In addition, Wang et al. showed that in order for Rap1 to activate the ERK pathway, Rap1 needs to be localized near the plasma membrane (*Wang et al., 2006*). These observations led us to hypothesize that a plasma membrane-associated pool of active Rap1 exists to regulate plasma membrane ERK activity.

The classical way to measure GTPase activation involves affinity capture of the active GTPase and subsequent immunodetection for the presence of the GTPase in the precipitated fraction. We reasoned that because the plasma membrane-associated pool of Rap1 may be relatively small compared to the cytosolic and perinuclear pools of Rap1, this classical method involving affinity-capture would not be suitable to test whether EGF induces Rap1 (*Wang et al., 2006*; *Obara et al., 2007*). Recently, O'Shaughnessy et al. reported novel dual-chain biosensors for both of the isoforms of Rap1, Rap1A and Rap1B (*O'Shaughnessy et al., 2019*). In these biosensors known as Rap1A/B-FLARE, either Rap1A or Rap1B is fused to the donor fluorophore, mCerulean3, on its N-terminus and the Rap1-binding domain (RBD) from RalGDS is fused to the acceptor fluorophore Ypet. Rap1 activation leads to intermolecular binding between Rap1 and RBD, thereby increasing FRET. Using these biosensors in PC12 cells (see *Figure 3—figure supplement 1B* for diagram of pathway including inhibitors and biosensors utilized throughout this study), we observed that EGF consistently induced increases in Y/C emission ratios, indicating activation of both Rap1A (25.4 ± 4.7%, n = 11) and Rap1B (26 ± 3.4%, n = 8); this effect was completely abrogated by either co-expression of the Rap1-specific GTPase-activating protein Rap1GAP, which selectively inactivates Rap1 by inducing the hydrolysis of the bound GTP to GDP (*Figure 3A,B*, red curves, *Figure 3—figure supplement 2*; *Brinkmann et al., 2002*; *Chen et al., 1997*), or by GGTI-298, a geranylgeranyltransferase I inhibitor that has been used to inhibit the processing, and thus action, of Rap1 (*Figure 3—figure supplement 3*; *Chenette and Der, 2011*; *Ke et al., 2019*; *McGuire et al., 1996*). We also tested Rap1 activation near the plasma membrane by using Total Internal Reflection Microscopy (TIRF) to monitor the translocation to the plasma membrane of a FP-tagged effector molecule that binds to GTP bound form of Rap1. For this assay, we utilized the RBD from diffusible, FP-tagged RalGDS (*Wang et al., 2006*). We also co-expressed a membrane-targeted variant of RFP, mCherry, in PC12 cells in order to aid in finding cells in TIRF microscopy and to control for changes in fluorescence intensity due to cell spreading and/or movement. In these cells, EGF induced an average maximum response of 64.8 ± 11% (n = 11) increase in ratio of EGFP to mCherry at the basal membrane, while control cells expressing EGFP showed no changes in EGFP fluorescence intensity at the membrane

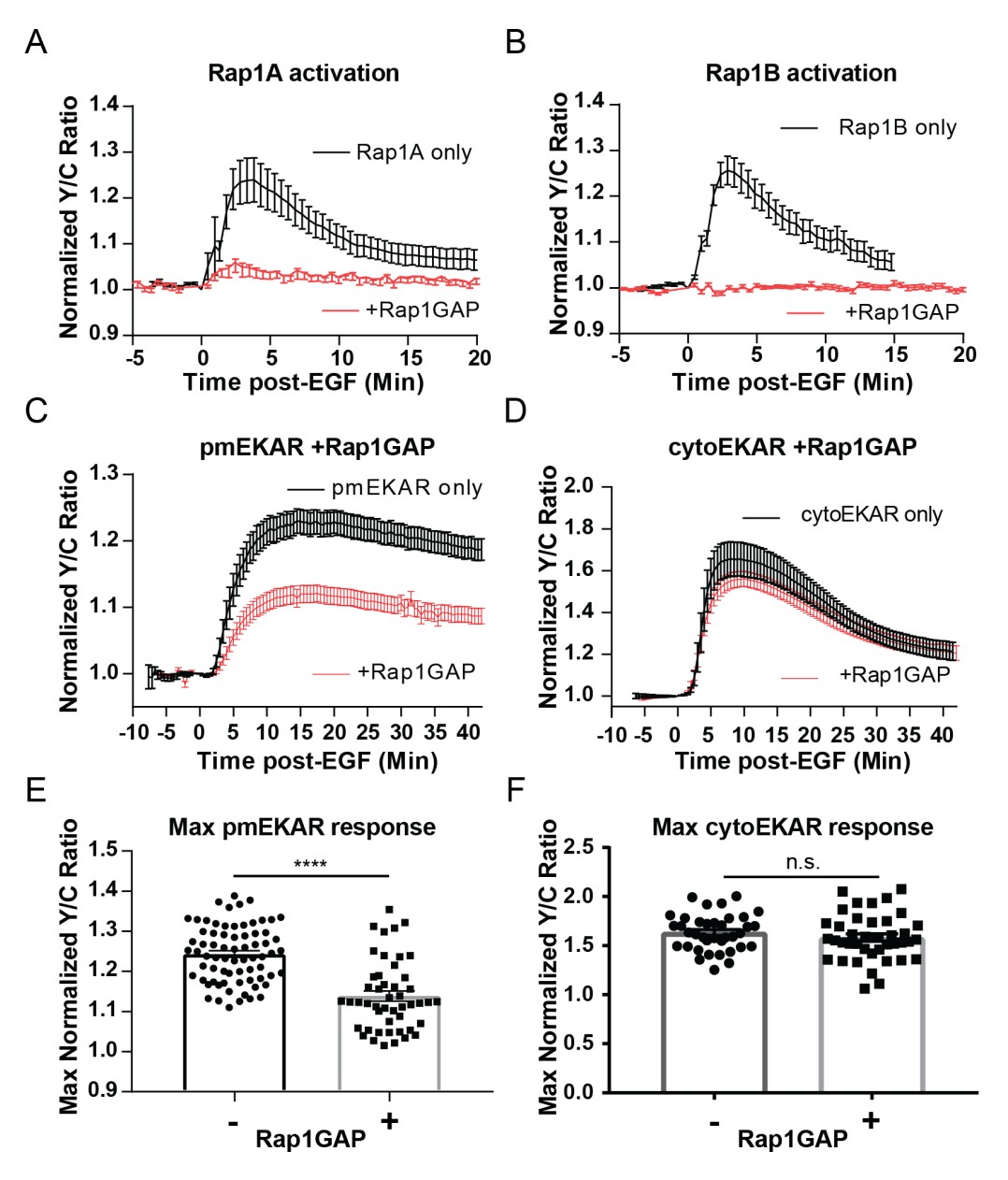

**Figure 3.** Rap1 GTPase regulates EGF-induced ERK activity at the plasma membrane. Cells expressing either (**A**) diffusible Rap1A FLARE (n = 11) or (**B**) diffusible Rap1B FLARE (n = 9) were treated with EGF at time 0, increase in yellow over cyan (Y/C) emission ratio indicates Rap1 activation. The observed increases in ratio are abrogated by co-expression of the Rap1-specific GAP Rap1GAP (red curves, n = 14, n = 15, respectively). (**C, E**) Rap1GAP expression significantly dampened pmEKAR4 response (n = 40). (**D, F**) Rap1GAP expression had no significant effect on cytosolic ERK activity (n = 40). (****p<0.0001, n.s. = not significant, calculated using student's t-test with Welch's correction) See also *Figure 3—figure supplements 1–6*.

The online version of this article includes the following source data and figure supplement(s) for figure 3:

**Source data 1.**

**Figure supplement 1.** Model of ERK and tested pathways.

**Figure supplement 2.** Rap1-FLARE responses of all replicates.

**Figure supplement 3.** Abrogation of Rap1 signaling via geranylgeranyl transferase inhibition significantly dampens plasma membrane ERK response.

**Figure supplement 4.** Ral-GDS translocation assay reveals EGF-mediated activation of Rap1 at the plasma membrane.

*Figure 3 continued on next page*

*Figure 3 continued*

**Figure supplement 5.** Dominant negative Rap1 isoforms on EGF-induced Rap1 and plasma membrane ERK responses.

**Figure supplement 6.** Effect of HRasN17 dominant negative expression on cytoEKAR4 and pmEKAR4 response to EGF in PC-12 cells.

(*Figure 3—figure supplement 4*). Furthermore, co-expression of Rap1GAP abolished the translocation of the RBD, further confirming the specificity of the response.

Having verified EGF induces Rap1 activation near the plasma membrane, we investigated how abrogation of Rap1 would affect plasma membrane localized ERK activity. We used several different approaches to inhibit Rap1. First, Rap1GAP-mediated inhibition of both Rap1A and Rap1B led to a dramatic dampening of EGF-induced ERK activity near the plasma membrane (*Figure 3C,E*, and *Figure 3—figure supplement 2*). However, Rap1GAP had no effect on cytosolic ERK activity (*Figure 3D,F*), indicating that the effect of Rap1GAP is plasma membrane specific. Secondly, we expressed Rap1A and Rap1B dominant-negative isoforms (Rap1A N17 and Rap1B N17) (*Maly et al., 1994*). While Rap1A N17 was not able to inhibit Rap1A activation, Rap1B N17 dramatically decreased Rap1B activation (*Figure 3—figure supplement 5B,D*) and led to a similar effect on pmEKAR response as Rap1GAP expression (*Figure 3—figure supplement 5F,G*). In addition, acute Rap1A inhibition by GGTI-298 also resulted in a drastically decreased pmEKAR4 response (*Figure 3—figure supplement 3*). These data indicate that Rap1 is critically involved in regulating plasma membrane ERK activity. However, we note that Rap1 inhibition does not completely abrogate plasma membrane ERK response. We thus used HRas-N17, a dominant negative form of H-Ras that inhibits all Ras isoforms (*Matallanas et al., 2003*), to probe how inhibition of Ras affects both cytosolic and plasma membrane localized ERK. We found that cytosolic ERK activity was completely abolished (*Figure 3—figure supplement 6A,B*). We also found that plasma membrane localized ERK activity was altered, but not abolished like the cytoEKAR4 response. Instead of an initial peak response approximately 10 min after EGF followed by a slow decay, we observed a slowly increasing plasma membrane ERK response (*Figure 3—figure supplement 6C,D*). Interestingly, the amplitude of pmEKAR4 at 40 min was not different between HRas-N17 expressing cells and control cells, indicating that Rap1, not Ras, is responsible for plasma membrane ERK activity at later time points. This mechanism is notably different from the canonical Ras-mediated activation of ERK, which appears to regulate the cytoplasmic and nuclear ERK activities.

## Plasma membrane ERK activity regulates cellular morphology and EGF-induced protrusion dynamics

After identifying a distinct pool of ERK with different temporal dynamics and pathway of activation than cytosolic or nuclear pools of ERK, we sought to identify the function of this pool of ERK near the membrane. We utilized an ERK monobody binder (EMBer) (*Mann et al., 2013*) that interacts with ERK and inhibits its activity but does not interact with JNK or p38, kinases from the same family as ERK. Using plasma membrane targeted EMBer tagged with mCherry (pmEMBer), we observed a significantly reduced ERK activity near the plasma membrane (*Figure 4A*) but not in the cytosol (*Figure 4B*) or nucleus (*Figure 4—figure supplement 1*). Thus, plasma membrane targeted EMBer could indeed be used to selectively inhibit the plasma membrane ERK activity. In cells expressing pmEMBer, we first observed an apparent change in cell morphology from primarily circular to primarily elliptical (*Figure 4C*). We quantified this morphological change by measuring the ratio of the minor to major axes of each cell, comparing between pmEMBer expressing cells and non-pmEMBer expressing cells (*Figure 4D*, Left panel). In cells without pmEMBer, 58% of imaged cells have a minor:major axes ratio greater than 0.8, indicating that the majority of cells were nearly perfectly round. Only ~8% of cells without pmEMBer had a ratio below 0.6. In contrast, 42% of pmEMBer expressing cells had a ratio below 0.6. This drastic change in cell morphology, which is dependent on pmEMBer expression at the plasma membrane, indicates that ERK activity near the membrane has an important role in controlling cell morphology.

In conjunction with the observed changes in morphology, we observed apparent changes in EGF-induced membrane protrusion dynamics. To quantitate these observations, we co-expressed either

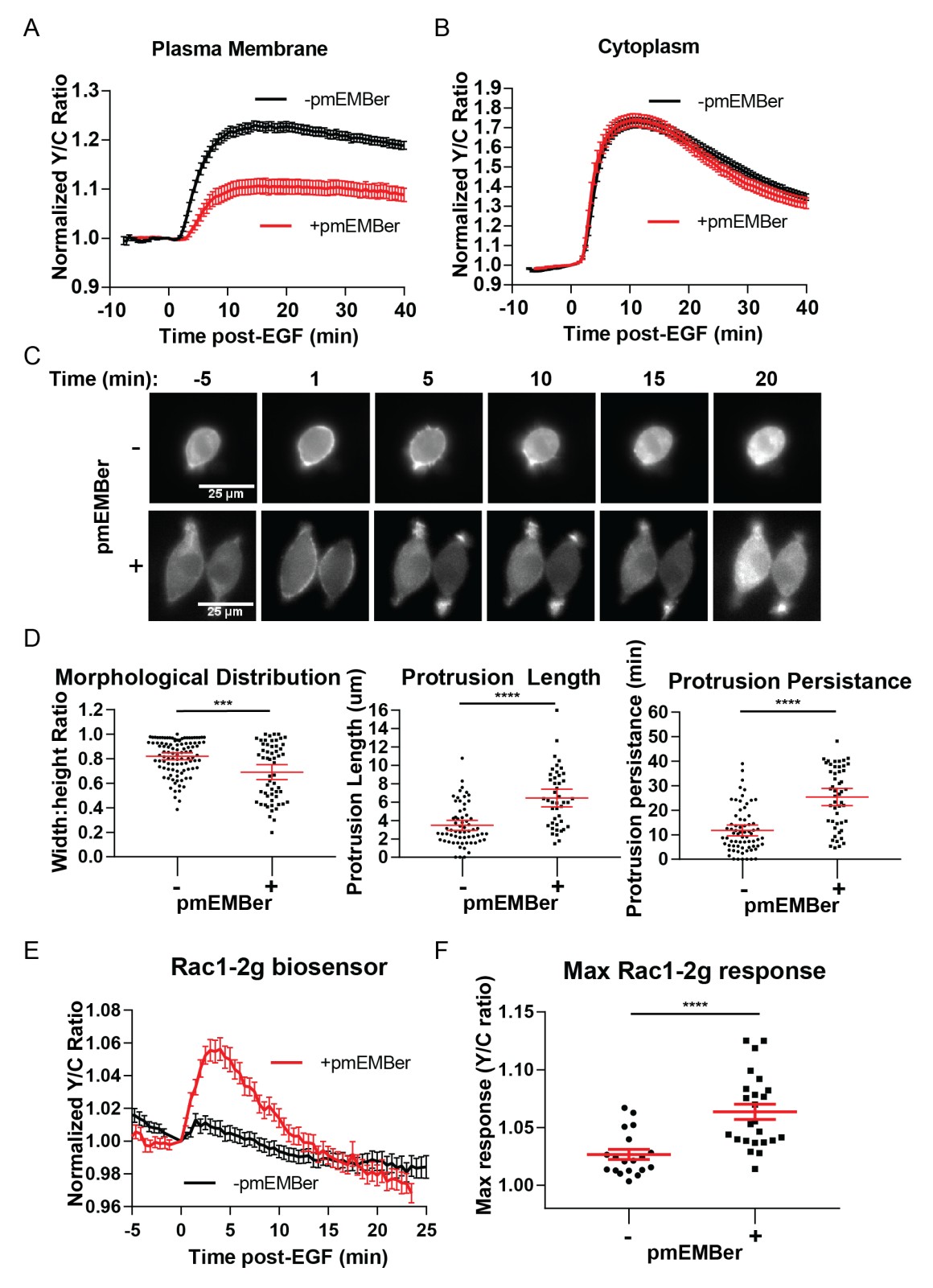

**Figure 4.** Inhibition of plasma membrane localized ERK activity alters cell morphology, EGF-induced protrusions, and Rac1 activation. (A) Effect on pmEKAR4 response to EGF in PC12 cells expressing a plasma membrane-targeted monobody EMBer7.1 (pmEMBer) (*Mann et al., 2013*) which binds to ERK and dampens ERK activity (n = 36 cells, red curve). (B) Effect on cytoEKAR4 response to EGF in PC12 cells expressing pmEMBer (n = 37 cells, red curve) (C) Representative images of cell morphology and protrusion dynamics in response to EGF in PC12 cells with (bottom row) or without (top

*Figure 4 continued on next page*

eLife Research article

Biochemistry and Chemical Biology | Cell Biology

Figure 4 continued

row) pmEMBer expression. (D) (Left-most panel) Major and minor axes of each cell was measured using ImageJ and the ratio of minor: major was calculated for each cell with and without pmEMBer expression. Kymographs generated from major axes of each cell plotting distance and time (*Figure 4—figure supplement 2*) were used to quantitate protrusion length in μm and protrusion persistence in min. (Middle and right panels) (n = 103 pmEMBer, n = 57 +pmEMBer, ***p=0.0002, ****p<0.0001) (E) Using Rac1-2G biosensor to measure Rac1 activity (*O'Shaughnessy et al., 2019*), cells either expressing pmEMBer (red curve, n = 24 cells) or pm-mCherry (black curve, n = 19 cells) were treated with EGF. (F) Quantitation of Max Rac1-2G response to EGF. (p<0.0001, Welch's t-test to correct for different variances.) See also *Figure 4—figure supplement 1*, *Figure 4—figure supplement 2*, *Figure 4—figure supplement 3*.

The online version of this article includes the following source data and figure supplement(s) for figure 4:

**Source data 1.**
**Figure supplement 1.** Effects of pmEMBer on ERK and PC-12 responses to EGF.
**Figure supplement 2.** Examples of cell protrusion measurements.
**Figure supplement 3.** Effect of pmEMBer on Rac1 and RhoA responses to EGF in PC12 cells.

pmEMBer or plasma membrane targeted mCherry with LifeAct-Venus (*Hertel et al., 2016*) to visualize actin dynamics. In each of these conditions, cells were treated with EGF and imaged every 30 s for up to 45 min after stimulation. To quantitate the differences in protrusion dynamics, lines were drawn across the major axis of each cell and a kymograph was generated to quantitate the spatial and temporal aspects of protrusion dynamics (*Figure 4—figure supplement 2*). Under control conditions, EGF induced relatively short-lived, evenly distributed small protrusions (*Figure 4D*, middle, right panels). However, under conditions of selective plasma membrane inhibition of ERK activity near the plasma membrane, cells exhibited longer and polarized protrusion dynamics (*Figure 4D*, middle, right panels). Specifically, in normal conditions, 28% of cells had protrusions longer than 5 μm and the protrusions of 25% of these cells lasted for longer than 15 min. In contrast, in plasma mebrane ERK (pmERK)-inhibited conditions, 70% of cells had protrusions longer than 5 μm and 75% of pmERK-inhibited cells had protrusions lasting longer than 15 min.

As described earlier, there are known ERK substrates near the membrane that could have the observed effect on cell morphology and protrusions, such as WASP, FAK, among others (*Ünal et al., 2017*). Furthermore, it is well established that the Rho family of GTPases, Rac1 and RhoA, are significant players in cell protrusion and migration processes. Importantly, Rac1 is a known substrate for ERK, and EGF-induced Rac1 phosphorylation by ERK in COS-7 cells was shown to lead to inhibition of Rac1 and its removal from the plasma membrane (*Tong et al., 2013*). Thus, we sought to examine how inhibition of plasma membrane ERK affected Rac1 and RhoA activity in PC12 cells.

In order to monitor Rac1 and RhoA activation, we took advantage of established single-chain Rac1 and RhoA biosensors: Rac1-2G (*Fritz et al., 2015*) and RhoA-DORA (*van Unen et al., 2015*). Rac1-2G consists of mTFP (variant of CFP) on the N-terminus, a circularly permutated variant of Venus, with the PAK effector binding domain in between the two fluorescent proteins and a full length Rac1 at the C-terminal end; upon Rac1 binding with GTP, the PAK effector domain will interact with Rac1-GTP, bringing the two fluorescent proteins into close proximity to increase FRET. The RhoA biosensor has a similar design, with a circularly permutated Rho-binding domain of PKN1 on the N-terminus, followed by cpVenus, a linker region, Cerulean, and then full-length RhoA. We expressed either of these biosensors in cells along with either plasma membrane targeted mCherry (control) or pmEMBer. Although we could not observe RhoA activity under either condition (*Figure 4—figure supplement 3C,D,E*), using Rac1-2G we did observe a robust 6.4 ± 0.7% increase in response to EGF (n = 19, control; n = 24, with pmEMBer) under pmERK-inhibited conditions but not under control conditions (*Figure 4E,F*), indicating endogenous pmERK inhibits Rac1 activation. This observation is consistent with previous reports in which ERK phosphorylation of Rac1 on T108 inhibits Rac1 and leads to relocalization of Rac1 away from the membrane (*Tong et al., 2013*) and indicates that plasma membrane ERK activity inhibits Rac1, which could contribute to the regulation of membrane protrusion dynamics.

## Discussion

Although ERK is ubiquitously utilized by a host of different cellular processes, the majority of studies on ERK focus on its primary role in proliferation and survival. Previous studies often focus on

investigating the extent of ERK phosphorylation and its effect on transcription factors in the nucleus. However, ERK is known to have hundreds of substrates localized in different subcellular compartments and play key roles in differentiation, migration, and others (*Roskoski, 2012*; *Ünal et al., 2017*; *Maik-Rachline et al., 2019*). Despite our knowledge of substrate localization, little is known about the regulation of ERK in specific subcellular compartments outside of the nucleus. This is largely due to the traditional method of using phospho-specific antibodies on whole lysates, which are equated, perhaps erroneously (*Keyes et al., 2017*; *Takahashi et al., 2012*), with full activity, and may mask subpopulations of ERK amidst the majority population. While we know that temporal dynamics plays a major role in determining cellular processes resulting from an extracellular cue, we have been unable to critically assess whether an individual extracellular signal can induce distinct temporal dynamics among two or more different subcellular compartments.

In this study, we established the use of fluorescent, FRET-based biosensors to study the spatio-temporal dynamics of ERK activity by targeting the biosensors to specific subcellular regions, particularly the plasma membrane, cytosol, and nucleus. Using this technique, we discovered that EGF simultaneously activates at least three divergent sub-populations of ERK with unique temporal dynamics from each other: the cytosol (*Figure 1C*), the Plasma membrane (*Figure 1D*), and the nucleus (*Figure 1—figure supplement 3C*). Indeed, the nuclear EKAR4 response exhibits the most transient of the targeted EKAR4 biosensors, followed by cytoEKAR4, with the pmEKAR4 exhibiting an apparent persistent response (*Figure 1D,E*). By using a membrane-permeable ERK inhibitor, we confirmed that the sustained pmEKAR4 signal was a direct result of continued ERK phosphorylation of the biosensor and not because the biosensor was unable to be dephosphorylated by a lack of phosphatases near the membrane (*Figure 2A*). Additionally, we utilized subcellular fractionation and immunoblotting to confirm the presence of sustained ERK phosphorylation in the plasma membrane fraction (*Figure 2B,C*).

To delineate into the mechanism by which ERK activity at the membrane is activated, we examined the upstream GTPase responsible for ERK activation and discovered a noncanonical Rap1-mediated pathway. Past studies have linked Rap1 to ERK signaling because Rap1 is able to interact with and activate B-Raf (*Takahashi et al., 2017*). However, this point has been controversial and it is not clear whether or not Rap1 actually activates ERK in a cellular environment, with studies giving disparate reports on whether EGF induces Rap1 activation in PC12 cells (*Wang et al., 2006*; *Zwartkruis et al., 1998*; *Kao et al., 2001*; *Wu et al., 2001*; *Bouschet et al., 2003*; *Li et al., 2006*). These contradicting observations are likely due to the fact that the traditional biochemical method for measuring enzyme activation may not capture transient or specific subcellularly localized activation (*Wang et al., 2006*). In this study, with the use of recently reported Rap1 biosensors (*Figure 3A–B*; *O'Shaughnessy et al., 2019*) and our live-cell TIRF-based EGFP-RalGDS translocation assay (*Figure 3—figure supplement 4*), we were able to show that, in PC12 cells, EGF does indeed activate Rap1 at the plasma membrane. Using Rap1GAP (*Figure 3C–F*), a dominant negative form of Rap1B (*Figure 3—figure supplement 5*), and the geranylgeranyl transferase I inhibitor (*Figure 3—figure supplement 3*), combined with our plasma membrane ERK activity reporter, we further showed that Rap1 is critically involved in regulating plasma membrane ERK activity but does not appear to contribute to the Ras-dependent cytosolic or nuclear ERK activities, whereas Ras appears to primarily contribute to the initial, but not sustained, component of plasma membrane ERK activity (*Figure 3—figure supplement 6*). Future studies will test the involvement of Rap1 GEFs and B-Raf in Rap1-mediated plasma membrane ERK activation.

Furthermore, our study provides a striking example where an individual extracellular signal, such as EGF, can temporally regulate subcellular ERK populations to control distinct cellular processes. We showed that inhibition of membrane-localized ERK activity potentiates EGF-induced Rac1 activation and leads to enhanced membrane protrusion dynamics. These data suggest that EGF, in addition to inducing cell proliferation, also stimulates plasma membrane ERK activity to suppress Rac1 activity to prevent unwanted membrane protrusions and polarization. The latter effect likely requires the sustained ERK activity at the plasma membrane. It is tempting to speculate that the ERK-suppressed morphology and protrusions are related to PC12 cell differentiation, due to the qualitative resemblance of shape and protrusion dynamics for pmEMBer-expressing cells compared to cells shortly after NGF treatment (*Robinson et al., 1998*). Thus, spatial and temporal dynamics are intricately linked to encode specific signaling information and regulate distinct cellular processes that result from individual extracellular signals such as EGF. With the genetically targeted molecular tools

for both visualization and manipulation of specific signaling events, our work also lays the foundation to enable future studies to further elucidate the mechanisms of regulation and functional roles of distinct subcellular populations of ERK.

# Materials and methods

## Key resources table

| Reagent type (species) or resource | Designation | Source or reference | Identifiers | Additional information |
|---|---|---|---|---|
| Chemical compound, drug | SCH772984 | selleckchem | S7101 | |
| Chemical compound, drug | U0126 | Sigma | U120 | |
| Chemical compound, drug | Epidermal Growth Factor (EGF) | Sigma | E9644 | |
| Antibody | Anti-phosphoERK1/2 rabbit, polyclonal | Cell Signalling Technology | Cat# 9102 RRID:AB_330744 | (1:1000) dilution |
| Antibody | Anti-ERK1/2 rabbit, polyclonal | Cell Signalling Technology | Cat# 9101 RRID:AB_331646 | (1:1000) dilution |
| Commercial assay or Kit | Lipofectamine-2000 | Invitrogen | 11668019 | |
| Recombinant DNA Reagent | pCDNA3 EKAR-EV | PMID:21976697 | | |
| Recombinant DNA Reagent | pTriEx4 Rap1A-FLARE | PMID:31444239 | | |
| Recombinant DNA Reagent | pTriEx4 Rap1B-FLARE | PMID:31444239 | | |
| Recombinant DNA Reagent | pcDNA3 Flag-Rap1GAP | PMID:23946483 | | |
| Recombinant DNA Reagent | pcDNA3 EMBer7.1 | PMID:23227961 | | |
| Cell Line (*Rattus norvegicus*) | PC-12 | PMID:1065897 | RRID:CVCL_0481 | |
| Cell Line (*Homo sapiens*) | HEK293-T | | RRID:CVCL_0063 | |
| Software, algorithm | FIJI | | RRID:SCR_014294 | |
| Software, algorithm | MetaFluor | | RRID:SCR_002285 | |
| Software, algorithm | GraphPad Prism | | RRID:SCR_002798 | |

## Reagents

ERK inhibitor SCH772984 was obtained from Sellek chemicals. ERK1/2 (Cell Signaling Technology Cat# 9102, RRID:AB_330744) and phospho-ERK1/2 (Cell Signaling Technology Cat# 9101, RRID:AB_331646) antibodies were from CST. Epidermal growth factor (EGF) was from Sigma-Aldrich.

## Plasmid generation and molecular biology

EKAR-EV was kindly provided by Michiyuki Matsuda. To generate EKAR4, locations of ECFP and Ypet were swapped by restriction digest. Subcellular targeting sequences were subcloned onto the C-terminal end of EKAR4 in between the EcoRI and XbaI restriction sites. The sequences include: for nuclear export sequence (5'-GAATTCCTGCCCCCCCCTGGAGCGCCTGACCCTGTAA-3'), for nuclear import sequence (5'-CCAAAAAAGAAGAGAaaggtggaagacgca-3'), and for plasma membrane localization (5'-AAGAAGAAGAAGAAGAGCAAGACCAAGTGCGTGATCATGTAA-3'.) The corresponding peptide sequences are LPPLERLTL, PKKKRKVEDA, and KKKKKSKTKCVIM, respectively. The full annotated sequence for cytoEKAR4 is included as *Supplementary file 1* and the sequence for pmE-KAR4 is included as *Supplementary file 2*. Flag-Rap1GAP, Rap1A-N17 and Rap1B-N17, and EGFP-

RalGDS was kindly provided by Philip Stork. Flag-Rap1GAP was PCR-purified out of the provided plasmid using the forward primer (5'- gggatctgtacgacgatgacgataagGATCCCATGATTGAGAAGA TGCAGGGAAGCAGGA-3') and reverse primer (5'- ACAGCCCAGCTGGGGCATGCCCTTGC TCACCATGTCGACAAAGGTACC-3'). The PCR product was then subcloned using Gibson assembly into pcDNA3 containing mCherry using the following primers: (5'- TCTGAGCAGCACA TGCCCCAGCTGGGCTGTGGTACCTTTGTCGACATGGTGAGCAAGGGC-3'), (5'-cttatcgtcatcgtcgta-cagatcccgacc-3'), (5'- ATTGCATCGCATTGTCTGAGTAGGTG-3'), and (5'-CGCTCTTTCTGGTCA TCCAGGCGAA-3'). Sequences for ERK monobody binders kindly provided by Sheldon Park. EMBer7.1 was subcloned into pRset containing the mCherry fluorescent protein between the BamHI and SacI sites such that EMBer7.1 would be tagged with mCherry on its C-terminal end. EMBer7.1-mCherry was then subcloned into pcDNA three after silent mutation of the internal BamHI site within EMBer7.1 between the BamHI and EcoRI sites. The plasma membrane targeting sequence was inserted on the C-terminal end of mCherry between the EcoRI and XbaI sites; the subsequent product was termed pmEMBer7.1. Creation of LifeAct-Venus was previously reported (Hertel et al., 2016). Rac1-2g in pTriEx4 was a gift from Oliver Pertz (Addgene plasmid # 66110), and Dora-RhoA was kindly provided by Yi Wu (University of Connecticut). Rap1A-FLARE and Rap1B-FLARE were kindly provided by Klaus Hahn (UNC-Chapel Hill).

## Cell culture and transfection

PC12 cells were grown in Dulbecco modified Eagle medium (DMEM) cell culture medium supplemented with 10% fetal bovine serum (FBS) and 5% donor horse serum (DHS) at 37°C with 5% $CO_2$. HEK293T cells were maintained using Dulbecco's Modified Eagle's Medium (DMEM) supplemented with 10% fetal bovine serum (FBS) and 1% penicillin/streptomycin. Cells were plated onto 35 mm glass-bottom dishes for imaging, 60 mm culture dishes for immunoblotting studies, or 15 cm culture dishes for fractionation studies. When necessary, cells were transfected at 50% to 60% confluence via Lipofectamine 2000 (Invitrogen) for 24–48 hr prior to imaging. Cells were recovered from frozen stocks, maintained for limited passages and screened regularly to confirm the absence of mycoplasma contamination using Hoechst staining.

## Cellular fractionation and immunoblotting

PC12 cells were washed 2x with ice cold 1X PBS and harvested in 1X PBS. Cells were centrifuged for 5 min at 5000 RPM at 4 °C, resuspended in 250 µL of 1X hypotonic lysis buffer (20 mM HEPES, pH 7.4, 10 mM NaCl, 3 mM MgCl2, Roche protease cocktail inhibitors and 1 mM PMSF), and incubated from 30 min to 1 hr to allow for cell lysis. Cells were passed through a 27 ½ G needle 10x to maximize lysis and centrifuged 2x at 5000 RPM to pellet nucleus. BCA quantification was performed and protein levels were normalized to the lowest protein concentration (2–2.5 mg/mL). Pellets were washed 2x and combined in 250 µL of hypotonic buffer. Supernatant was ultracentrifuged at 52,000 RPM for 1 hr at 4 °C. Supernatant was saved as the cytoplasmic fraction. Membrane pellet was resuspended in 250 µL of hypotonic buffer and passed sample through a 27 ½ G syringe 10x, and centrifuged for 45 min at 52,000 RPM at 4 °C. Supernatant was combined with the cytoplasmic for a total of 500 µL. Membrane pellet was resuspended in 250 µL of hypotonic buffer, combined with the nuclear fractions with membrane fractions and solubilized membrane by adding NP-40 at a final concentration of 0.5%. Lysates were centrifuged 2x at 5000 RPM to pellet nucleus where your supernatant is the membrane fraction. The pellets were combined in 500 mL of hypotonic and sonicated to solubilize. Samples were prepared with LSB+DTT and assessed via SDS-PAGE. 9% gels were resolved and transferred to PVDF membranes. Membranes were blocked for 1 hr in 4% BSA and probed for rbαNa$^+$/K ATPase$^+$ (1:1000), msαP84 (1:2000), msαGAPDH (1:50,000), p-ERK (1:1000) and total ERK (1:1000). Phospho-ERK was normalized to total ERK within its corresponding fraction. Fraction of maximal response was obtained and analyzed repeats in Prism via Two-way ANOVA.

## Cellular imaging and analysis

### Epifluorescence imaging

Cells were washed twice with Hank's balanced salt solution (HBSS, Gibco) and incubated at 37°C for 30 min prior to imaging. Cells were subsequently imaged in HBSS in the dark at 37°C. Epidermal growth factor (EGF; Sigma-Aldrich) and SCH772984 (Selleck) were added as indicated; a minimum

of three biological replicates (individual dish) were collected for each condition. Images were acquired on a Zeiss AxioObserver Z1 microscope (Carl Zeiss) equipped with a 40×/1.3NA objective and a Photometrics Evolve 512 EMCCD (Photometrics) controlled by METAFLUOR 7.7 software (Molecular Devices). Dual cyan/yellow emission ratio imaging was performed using a 420DF20 excitation filter, a 450DRLP dichroic mirror and two emission filters (475DF40 for CFP and 535DF25 for YFP). All filter sets were alternated by a Lambda 10–2 filter-changer (Sutter Instruments). Exposure times ranged between 50 and 500 ms, with the EM gain set from 10 to 50, and images were acquired every 30 s. Raw fluorescence images were corrected by subtracting the background fluorescence intensity of a cell-free region from the emission intensities of biosensor-expressing cells. Yellow/cyan emission ratios were then calculated at each time point. The resulting time courses were normalized by dividing the ratio at each time point by the basal value at time zero (R/R0), which was defined as the time point immediately preceding drug addition. Maximum ratio (ΔR/R) changes were calculated as (Rmax−Rmin)/Rmin, where Rmax and Rmin are the maximum and minimum ratio value recorded after stimulation, respectively. Metric for sustained/transient comparisons was calculated as a ratio of remaining C/Y emission ratio 40 min after EGF stimulation to the maximum ratio response $(R_{40}-R_0)/(R_{max}-R_0)$. Where $R_{40}$ is the ratio at 40 min post-EGF treatment, $R_0$ is the point preceding EGF stimulation, and $R_{max}$ is the maximum ratio value recorded after stimulation. Graphs were plotted using GraphPad Prism 7 (GraphPad Software). Sample size reported as number of individual cells.

### Morphology and protrusion analysis

Morphology was measured using ImageJ software to measure the distance, in microns, of the major and minor axes of each cell, and the ratio of the minor:major axes was compared between cells with versus without pmEMBer expression. Protrusion dynamics was measured using ImageJ software by creating a line across the major axis of each cell, and subsequently creating a kymograph from the stack of time course images. For the kymograph, the x-axis indicated distance (microns) and the y-axis indicated time (30 s per pixel). Using the kymographs, length in distance and time of each protrusion was measured and compared between conditions (*Figure 4—figure supplement 2*).

### Total internal reflection microscopy (TIRF)

Images were acquired on an Olympus IX-71 inverted microscope under a 150 × objective. An optical system with a 488 and 561 nm solid-state sapphire, coherent excitation laser and an EMCCD camera (iXon DU897E, Andor Technology) was used to examine the samples (emission, ZET488/561 m, Chroma Technology). Images in green (488 nm) and red (561 nm) channels were taken every 30 s before and after 100 ng/ml EGF stimulation. Resulting time course images were analyzed using ImageJ software. Intensity of each cell was measured throughout time course, ratio of GFP/RFP was taken to account for cell spreading that sometimes occurs after EGF treatment.

### Statistical analysis

Statistics were calculated using Graphpad Prism seven software. When comparing average metrics of single cell curves, outliers were identified using ROUT with Q + 1%. Reported average is the mean ± SEM. Experiments comparing multiple conditions were compared were analyzed using Ordinary one-way ANOVA with multiple comparisons. Situations where only two conditions were compared were analyzed by an unpaired t-test, with Welch's correction if variances were shown to be different between the conditions. Statistical significance was assessed using $p < 0.05$ as a cutoff value. In combining single cell data, cells with clear unusual behavior (apoptotic blebbing) were cut from analysis, along with cells with extremely high or low start ratios or unusually bright or dim fluorescence (determined individually per condition by calculating Pearson's correlation coefficient between response metrics and Ratio or fluorescent starting conditions. This analysis was done to remove cells from analysis in which extremely high or low expression of the biosensor affected cell behavior; cells were cut from each condition until Pearson's correlation coefficient was below 0.5.

## Acknowledgements

We thank E Greenwald, G Mo, JF Zhang, and S Mehta for insightful discussions and for microscopy training. We thank D Schmidt for reviewing manuscript before submission. We thank P Stork for plasmids encoding Rap1A-N17, Rap1B-N17, EGFP-Rap1GAP, and EGFP-RalGDS. We also thank M Matsuda for EKAR-EV, S Park for EMBer monobody plasmids, Y Wu for RhoA-DORA biosensor, and O Pertz for Rac1-2G biosensor (Addgene # 66110). We also thank K Hahn for providing Rap1A and Rap1B biosensors.

## Additional information

### Funding

| Funder | Grant reference number | Author |
| --- | --- | --- |
| National Cancer Institute | R35 CA197622 | Jin Zhang |
| National Institute of Diabetes and Digestive and Kidney Diseases | R01 DK073368 | Jin Zhang |
| National Institute of General Medical Sciences | K12GM068524 | JoAnn Trejo |

The funders had no role in study design, data collection and interpretation, or the decision to submit the work for publication.

### Author contributions

Jeremiah Keyes, Conceptualization, Data curation, Formal analysis, Supervision, Validation, Investigation, Visualization, Methodology, Writing - original draft, Writing - review and editing; Ambhighainath Ganesan, Conceptualization, Data curation, Formal analysis, Investigation, Methodology; Olivia Molinar-Inglis, Data curation, Formal analysis, Investigation, Visualization, Methodology; Archer Hamidzadeh, Jinfan Zhang, Megan Ling, Data curation, Investigation; JoAnn Trejo, Resources, Supervision, Funding acquisition, Validation; Andre Levchenko, Conceptualization, Resources, Supervision, Validation, Project administration; Jin Zhang, Conceptualization, Resources, Supervision, Funding acquisition, Validation, Visualization, Writing - original draft, Project administration, Writing - review and editing

### Author ORCIDs

Jeremiah Keyes (iD) https://orcid.org/0000-0003-4253-6834
Jin Zhang (iD) https://orcid.org/0000-0001-7145-7823

### Decision letter and Author response

Decision letter https://doi.org/10.7554/eLife.57410.sa1
Author response https://doi.org/10.7554/eLife.57410.sa2

## Additional files

### Supplementary files

- Supplementary file 1. The full annotated sequence for cytoEKAR4.
- Supplementary file 2. The sequence for pmEKAR4.

### Data availability

All data generated and analyzed in this study are included in manuscript and figures.

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
