## [Decision Letter]

**Acceptance summary:**

This study examines the temporal and spatial dynamics of ERK signaling using a newly developed sensor. The authors demonstrate that cell surface signaling to ERK is mediated initially by both Ras and Rap pathways, but long-term ERK activation is mediated by the Rap pathway. This finding provides novel insight into ERK signaling dynamics and clarifies the relative roles of the Ras and Rap pathways.

**Decision letter after peer review:**

[Editors’ note: the authors submitted for reconsideration following the decision after peer review. What follows is the decision letter after the first round of review.]

Thank you for submitting your work entitled "Signaling diversity enabled by Rap1-regulated plasma membrane ERK with distinct temporal dynamics" for consideration by *eLife*. Your article has been reviewed by three peer reviewers, and the evaluation has been overseen by a Reviewing Editor and Senior Editor. The reviewers have opted to remain anonymous.

Our decision has been reached after consultation among the reviewers. Based on these discussions and the individual reviews below, the reviewers concluded that revision of your manuscript would require more than 2 months, and thus we cannot formally invite a revised version according to *eLife* policy. Nevertheless, the reviewers remain interested in your study and are willing to consider a new version of this manuscript that addresses the points raised during the review.

This manuscript describes a new probe for ERK activity (EKAR4) that is employed to probe ERK signaling in live cells. A focus of the analysis is a demonstration that cell surface ERK activity is dependent upon the Rap1 signaling pathway. These are elegant and interesting studies. However, further characterization of EKAR4 was requested by the reviewers and additional data to support the conclusions concerning the role of Rap1 are required. These concerns are elaborated in the appended reviews.

Reviewer #1:

This is an interesting study that describes the use of an improved biosensor for ERK activity (EKAR4) to investigate the spatiotemporal dynamics of ERK activity in response to treatment of PC12 cells with EGF. Many of the observations do not break new ground. However, the analysis of the role of Rap1 in cell surface activation of ERK does contribute to an unresolved area of previous research. The authors present data indicating the cell surface ERK activity is sustained and that this depends on Rap1.

The authors show that Rap1 is activated by EGF and that both RapGAP and dnRap1 suppresses cell surface activation of ERK. The function of cell surface ERK activation was probed using a targeted ERK monobody to demonstrate a role in membrane protrusion that may be mediated by a Rac1-dependent process. The analysis presented uses elegant optical probes and is interesting. However, the depth of mechanistic insight is limited – thus is a problem for a controversial area of research. For example, a caveat on the interpretation of the RapGAP and dnRap1 studies is that the transfection protocol may cause long-term changes in EGFR signaling due to trafficking or some other EGFR signaling property. It is therefore not certain that Rap1 mediates the effects of EGF on cell surface ERK activity rather than some indirect action of Rap1 on EGFR signaling. Control studies are needed. First, does blocking the Ras pathway prevent cell surface ERK activation? Presumably, if Rap1 (and not Ras) is required for cell surface ERK activation, then blocking Ras would not alter cell surface ERK activation. Second, does acute Rap1-deficiency prevent cell surface ERK activation – for example using induced Rap1 protein degradation. These studies are required to strengthen the conclusions of the current study and are important because of the existing controversy in this field.

Reviewer #2:

General assessment of work:

This manuscript reports a new genetically encoded FRET sensor for ERK, named EKAR4, which has enhanced dynamic range compared to previous EKAR versions. The EKAR4 construct targeted to the plasma membrane is used to reveal novel evidence for sustained activation of membrane-localized ERK. The sensor is used in combination with inhibitors of Rap1 (Rap1-GAP and DN-Rap1) to address a previous claim that Rap1 is needed for ERK activation, which has been controversial and difficult to corroborate. The results show convincingly that plasma membrane associated ERK is partially dependent on Rap1-GTP. A plasma membrane targeted ERK monobody (EMBer) is used to demonstrate the importance of plasma membrane ERK on cell morphology and membrane protrusions.

This manuscript represents an important contribution to the signaling field, by describing a new probe for ERK signaling and using it to address long held ambiguities about the activation and function of plasma membrane associated kinase. The experiments are carefully done and the results will be of broad interest.

A number of points are raised to improve accuracy and clarity.

Detailed comments:

1) Figure 1—figure supplement 1, Results paragraph one. It is unclear how the figure shows a 74.5% increase in Y/C emission ratio. Needed is a panel showing the direct comparison of EKAR4 against the previous generation EKARs, to document performance of the new sensor.

2) Figure 2B,C. The Western blot in panel B looks as if phosphoERK at the membrane nearly returns to baseline after 40 min, while phosphoERK in the cytosol remains higher than baseline at 40 min. This seems inconsistent with the quantitative data in panel C.

3) Figure 3—figure supplement 2 panel F. The experiments showing that Rap1GAP does not affect cytoEKAR (EKAR4-NES?) were averaged from n=15, whereas pmEKAR experiments were averaged from n=40. Data for pmEKAR with Rap1GAP showed a wide range of responses, where many time courses overlapped those of control. Concievably cytoEKAR might also show the same, if assayed to the same depth. Alternatively, the Rap1BN17 experiments should be performed with EKAR4-NES to confirm that the cytoplasmic sensor is unresponsive to Rap1 inhibition.

Reviewer #3:

Keyes et al. describe a refinement of the EKAR genetically encoded biosensor for Extracellular regulated kinase (ERK) activity. Using similar technology that was used to engineer the original biosensor, the authors have improved sensitivity as evidenced by a 74 % increase in the Y/C emission ratio. More direct comparison to conventional methods of evaluating ERK activation should be also included. While there is no doubt that the development of this new reporter is valuable, the utilization of EKAR4 in this study does not reveal any substantial biological advances.

1) Experiments in Figure 1 suggest that the signal to noise ratio of the membrane associated EKAR4 is less than cytoplasmic. Control experiments need to be included to ensure that the sustained activity metric presented in Figure 1E is really significant. Cytoplasm is spelt wrong in the title of Figure 1C.

2) Additional drug experiments with MEK inhibitors should be added to the data in Figure 2A. Targeting this upstream kinase in the RAF-MEK-ERK pathway is a more standard method of pathway inhibition. These experiments could be used to further justify the development of another ERAR and establish if there are any MEK independent effects.

3) The usage of multiple fluorescent biosensors to detect events upstream ERK signaling such as Rap 1 activations makes this work harder to follow. Also, can the authors show that activation of Eapc1 is involved in this process by treating with selective cAMP analog activators.

4) It is stated in the discussion "we found that ERK activation in the nucleus was transient, slightly more sustained in the cytoplasm and drastically more sustained near the plasma membrane". This seems a rather obvious statement since the agonist used to stimulate ERK activation is delivered from the plasma membrane. The authors need to provide more compelling evidence and rational for this conclusion.

[Editors’ note: further revisions were suggested prior to acceptance, as described below.]

Thank you for submitting your article "Signaling Diversity Enabled by Rap1-Regulated Plasma Membrane ERK with Distinct Temporal Dynamics" for consideration by *eLife*. Your article has been reviewed by three peer reviewers, and the evaluation has been overseen by a Reviewing Editor and Philip Cole as the Senior Editor. The reviewers have opted to remain anonymous.

The reviewers have discussed the reviews with one another and the Reviewing Editor has drafted this decision to help you prepare a revised submission.

This manuscript represents an important contribution to the signaling field, by describing a new probe for ERK signaling and using it to address long-held ambiguities about the activation and function of plasma membrane associated kinase. The experiments are carefully done and the results will be of broad interest. The responses to the reviews were thorough and comprehensive. The findings of the revised manuscript are significant and at a level appropriate for this journal.

Two minor issues should be addressed by the authors before the paper is published.

1) The new supplemental data presented in Figure 3—figure supplement 6 appears to be noisy (particularly figure D). It might help interpretation if the analyses used the same scale for the quantification of the Y/C ratio as figure C. There is a similar issue, but to a lesser extent, with Figure 3—figure supplement 6 (A and C).

2) New immunoblot data presented in Figure 1—figure supplement 5. Please confirm in the figure legend that this figure panel represents the same extract examined in separate blots for ERK and pERK and that this is why two tubulin loading controls are presented. Otherwise, please explain.

---

## [Author Response]

[Editors’ note: the authors resubmitted a revised version of the paper for consideration. What follows is the authors’ response to the first round of review.]

Reviewer #1:This is an interesting study that describes the use of an improved biosensor for ERK activity (EKAR4) to investigate the spatiotemporal dynamics of ERK activity in response to treatment of PC12 cells with EGF. Many of the observations do not break new ground. However, the analysis of the role of Rap1 in cell surface activation of ERK does contribute to an unresolved area of previous research. The authors present data indicating the cell surface ERK activity is sustained and that this depends on Rap1.The authors show that Rap1 is activated by EGF and that both RapGAP and dnRap1 suppresses cell surface activation of ERK. The function of cell surface ERK activation was probed using a targeted ERK monobody to demonstrate a role in membrane protrusion that may be mediated by a Rac1-dependent process. The analysis presented uses elegant optical probes and is interesting. However, the depth of mechanistic insight is limited – thus is a problem for a controversial area of research. For example, a caveat on the interpretation of the RapGAP and dnRap1 studies is that the transfection protocol may cause long-term changes in EGFR signaling due to trafficking or some other EGFR signaling property. It is therefore not certain that Rap1 mediates the effects of EGF on cell surface ERK activity rather than some indirect action of Rap1 on EGFR signaling. Control studies are needed. First, does blocking the Ras pathway prevent cell surface ERK activation? Presumably, if Rap1 (and not Ras) is required for cell surface ERK activation, then blocking Ras would not alter cell surface ERK activation. Second, does acute Rap1-deficiency prevent cell surface ERK activation – for example using induced Rap1 protein degradation. These studies are required to strengthen the conclusions of the current study and are important because of the existing controversy in this field.

We thank reviewer 1 for the detailed and thoughtful comments. We have performed additional experiments, which have led to a more complete understanding of the roles of Ras and Rap1 in regulating ERK.

As suggested by the reviewers, we performed additional experiments to test how inhibition of Ras, via use of HRas-17N dominant negative overexpression, effects ERK responses. We found Ras inhibition affects both pmEKAR4 and cytoEKAR4 responses. As shown in Figure 3—figure supplement 6, HRasN17 expression completely abolishes cytosolic ERK response and decreased the plasma membrane ERK response by 57% at normal peak signal (~10-15 min) but 0% at 40 minutes post EGF, suggesting that Ras is not required for the sustained ERK response at the plasma membrane but required for regulating both cytosolic ERK and the initial activation of plasma membrane ERK. Indeed, as shown in Figure 3C, Rap1GAP expression does not completely abrogate pmEKAR4 response. Given that expression of Rap1GAP was shown to abolish Rap1 activation (Figure 3A and 3B), this data, combined with Figure 3 and Figure 3—figure supplement 6, suggests that both Rap1 and Ras contribute to the plasma membrane ERK response but Rap1, not Ras, is responsible for plasma membrane ERK activity at later time points. We discuss these results in paragraph two of subsection “Rap1 GTPase significantly contributes to plasma membrane ERK activity” and Discussion paragraph three.

We tested different approaches to achieve acute inhibition of Rap1. We attempted to create a cell line where we could more acutely inhibit the Rap1 pathway using either a targeted degradation system or a cell line in which we could induce the overexpression of Rap1GAP. However, we were unsuccessful in creating a stable PC12 cell line expressing either of these constructs due to significant cell death after viral transduction. We thus utilized GGTI-298, a geranylgeranyltransferase I inhibitor that has been used to inhibit the processing and thus action of Rap1 (Chenette and Der, 2011; Ke et al., 2019; McGuire et al., 1996). Because Rap1, but not Ras, is geranylgeranylated, GGTI-298 is used as a semi-specific inhibitor for Rap1 (Chenette and Der, 2011). We found that 30 μM pretreatment of GGTI-298 150 minutes before EGF treatment effectively abrogated Rap1 signaling as measured using the Rap1A biosensor (Figure 3—figure supplement 3A, C, E) and had a dramatic effect on pmEKAR4 response (Figure 3—figure supplement 3B, D, F). While we acknowledge that GGTI-298 could have off target effects, we noted that 150-min pretreatment with GGTI-298 leads to a similar pmEKAR4 response as Rap1GAP overexpression and included this data as a supporting piece of evidence for the role of Rap1 in regulating plasma membrane ERK. We include discussion using GGTI-298 in subsection “Rap1 GTPase significantly contributes to plasma membrane ERK activity”.

Reviewer #2:General assessment of work:This manuscript reports a new genetically encoded FRET sensor for ERK, named EKAR4, which has enhanced dynamic range compared to previous EKAR versions. The EKAR4 construct targeted to the plasma membrane is used to reveal novel evidence for sustained activation of membrane-localized ERK. The sensor is used in combination with inhibitors of Rap1 (Rap1-GAP and DN-Rap1) to address a previous claim that Rap1 is needed for ERK activation, which has been controversial and difficult to corroborate. The results show convincingly that plasma membrane associated ERK is partially dependent on Rap1-GTP. A plasma membrane targeted ERK monobody (EMBer) is used to demonstrate the importance of plasma membrane ERK on cell morphology and membrane protrusions.This manuscript represents an important contribution to the signaling field, by describing a new probe for ERK signaling and using it to address long held ambiguities about the activation and function of plasma membrane associated kinase. The experiments are carefully done and the results will be of broad interest.A number of points are raised to improve accuracy and clarity.Detailed comments:1) Figure 1—figure supplement 1, . Results paragraph one. It is unclear how the figure shows a 74.5% increase in Y/C emission ratio. Needed is a panel showing the direct comparison of EKAR4 against the previous generation EKARs, to document performance of the new sensor.

Per reviewer 2’s request, we obtained EKAR-EV (Komatsu et al., 2011), EKAR2G1, (Fritz et al., 2013), and EKAR 3 (Sparta et al., 2015), and the yet unpublished EKAR3.5 from John Albeck (UC Davis), which is an improved version of EKAR3. We used HEK-293T cells to directly compare the dynamic ranges of these different EKARs. We found that the dynamic range of EKAR4 is significantly improved over previous generations of EKAR (Author response image 1). We were unable to express EKAR 3, and because EKAR3.5 is yet unpublished, we did not include it in Figure 1—figure supplement 1, which shows the significant improvement in dynamic range of EKAR4 over EKAR-EV and EKAR2G1. We discuss these results in paragraph one of the Results section.

**Author response image 1. sa2fig1:** EKAR4 exhibits improved dynamic range over existing EKAR constructs. A) Average line traces of EKAR responses to EGF in Hek-293T cells. B) Quantification of max amplitude responses from each respective EKAR. (**** p<0.0001 using one-way ordinary ANOVA, n=20 EKAR-EV and EKAR2G1, n=18 EKAR3.5, n=29 EKAR4.).

2) Figure 2B,C. The Western blot in panel B looks as if phosphoERK at the membrane nearly returns to baseline after 40 min, while phosphoERK in the cytosol remains higher than baseline at 40 min. This seems inconsistent with the quantitative data in panel C.

Per reviewer 1 and 2’s concerns, we reviewed and repeated the fractionation and western blots and replaced the previous representative image with a replicate whose image is more representative of the quantified data.

3) Figure 3—figure supplement 2 panel F. The experiments showing that Rap1GAP does not affect cytoEKAR (EKAR4-NES?) were averaged from n=15, whereas pmEKAR experiments were averaged from n=40. Data for pmEKAR with Rap1GAP showed a wide range of responses, where many time courses overlapped those of control. Concievably cytoEKAR might also show the same, if assayed to the same depth. Alternatively, the Rap1BN17 experiments should be performed with EKAR4-NES to confirm that the cytoplasmic sensor is unresponsive to Rap1 inhibition.

We are thankful for reviewer 2’s thoughtful consideration of replicate numbers and the variation in the pmEKAR response in Figure 3—figure supplement 2E. We have repeated the cytoEKAR4 experiment with Rap1GAP expression to reach the same N number (n=40) and concluded that Rap1GAP has no significant effect on cytosolic EKAR response. Changes are reflected in Figure 3 D, F, and Figure 3—figure supplement 2F. (Note the change in Figure supplement number due to addition of the diagram figure as Figure 3—figure supplement 1).

Reviewer #3:Keyes et al. describe a refinement of the EKAR genetically encoded biosensor for Extracellular regulated kinase (ERK) activity. Using similar technology that was used to engineer the original biosensor, the authors have improved sensitivity as evidenced by a 74 % increase in the Y/C emission ratio. More direct comparison to conventional methods of evaluating ERK activation should be also included. While there is no doubt that the development of this new reporter is valuable, the utilization of EKAR4 in this study does not reveal any substantial biological advances.

Per reviewer request, we included immunoblot time course data in Figure 1—figure supplement 5. The immunoblots were from whole cell lysate, which is the more traditional method of measuring ERK activation.

In this time course, we observe a dramatic increase in phosphorylated ERK 7.5 minutes after EGF treatment, followed by a dramatic decrease by 15 minutes. It is interesting that after 15 minutes we see a slight increase in phospho-ERK at the 30- and 40-minute time points. The activation profile is broadly similar to the EKAR response. The decrease in phospho-ERK is faster than the EKAR signal likely because the EKAR biosensor signal dynamics is dependent on both ERK and phosphatase activities. However, it is important to note that measurements of ERK phosphorylation is an indirect measure of ERK activity and that the phospho-ERK immunoblot is a measure of phosphorylated ERK from all compartments of all cells. We include discussion of this in the manuscript in paragraph two of the Results section.

1) Experiments in Figure 1 suggest that the signal to noise ratio of the membrane associated EKAR4 is less than cytoplasmic. Control experiments need to be included to ensure that the sustained activity metric presented in Figure 1E is really significant. Cytoplasm is spelt wrong in the title of Figure 1C.

It is true that pmEKAR4 has a lower dynamic range than cytoEKAR4. We thus included a new supplemental figure (Figure 1—figure supplement 4) in which we show no significant correlation between amplitude and SAM40, indicating that the lower amplitude in pmEKAR does not affect the SAM40 measurement.

We also fixed the spelling in Figure 1C.

2) Additional drug experiments with MEK inhibitors should be added to the data in Figure 2A. Targeting this upstream kinase in the RAF-MEK-ERK pathway is a more standard method of pathway inhibition. These experiments could be used to further justify the development of another ERAR and establish if there are any MEK independent effects.

We chose to use an ERK inhibitor to determine if ERK is still actively phosphorylating pmEKAR4 at a given time. But indeed MEK inhibition is a more widely used method to inhibit the pathway. Per reviewer request, we have included data using MEK inhibitor U0126 as a supplemental figure (Figure 2—figure supplement 2) and including brief discussion of the results in paragraph four of the Results section.

3) The usage of multiple fluorescent biosensors to detect events upstream ERK signaling such as Rap 1 activations makes this work harder to follow. Also, can the authors show that activation of Eapc1 is involved in this process by treating with selective cAMP analog activators.

To make it easier to follow, we have now included a supplementary figure (Figure 3—figure supplement 1 to illustrate the pathway and the various biosensors and inhibitors used in this study, referred to in subsection “Rap1 GTPase significantly contributes to plasma membrane ERK activity”.

We appreciate reviewer 3’s question concerning the involvement of Epac. Based on the literature (Liu et al., 2010; Obara et al., 2007; Wang et al., 2006), we believe that C3G is a more likely candidate than Epac for the EGF-induced activation of Rap1 in this system owing to the localization of Epac away from the membrane. However, investigation of the mechanism of Rap1 activation, which involves specific inhibition of different GEFs, we feel, is beyond the scope of this current manuscript and more appropriately aligned with ongoing follow up studies, which we mention in paragraph three of the Discussion.

4) It is stated in the discussion "we found that ERK activation in the nucleus was transient, slightly more sustained in the cytoplasm and drastically more sustained near the plasma membrane". This seems a rather obvious statement since the agonist used to stimulate ERK activation is delivered from the plasma membrane. The authors need to provide more compelling evidence and rational for this conclusion.

To address reviewer 3’s concern, we modified the wording in paragraph two of the Discussion to more accurately reflect our observations on the apparent transience or persistence of ERK activity using the targeted EKAR4 biosensors. We also slightly modified the wording in the same paragraph to emphasize how experiments in Figure 2 support the stated conclusion that ERK phosphorylation and activity is sustained near the plasma membrane.

[Editors’ note: what follows is the authors’ response to the second round of review.]

Two minor issues should be addressed by the authors before the paper is published.1) The new supplemental data presented in Figure 3—figure supplement 6 appears to be noisy (particularly figure D). It might help interpretation if the analyses used the same scale for the quantification of the Y/C ratio as figure C. There is a similar issue, but to a lesser extent, with Figure 3—figure supplement 6 (A and C).

We modified Figure 3—figure supplement 6 such that Panels A and B have the same scale on the Y-axis, and Panels C and D have the same Y-axis scale as each other.

2) New immunoblot data presented in Figure 1—figure supplement 5. Please confirm in the figure legend that this figure panel represents the same extract examined in separate blots for ERK and pERK and that this is why two tubulin loading controls are presented. Otherwise, please explain.

We thank the reviewers and editors for aiding in clarifying this supplemental figure. There were two sets of immunoblots, pERK1/2 + tubulin (top) and ERK1/2 + tubulin (bottom) that are from separately run gels. As indicated in the revised figure, the columns represent the two biological replicates, and the samples in a given column are from the same set of lysates. Each tubulin blot is a loading control for the associated ERK immunoblot. We clarified this in the figure legend and modified the figure itself to clarify the reason for the two tubulin blots.